# A metal-supported single-atom catalytic site enables carbon dioxide hydrogenation

Sung-Fu Hung [1,2,4], Aoni Xu[1,4], Xue Wang [1,4], Fengwang Li [1,4], Shao-Hui Hsu[3], Yuhang Li [1], Joshua Wicks [1], Eduardo González Cervantes[1], Armin Sedighian Rasouli[1], Yuguang C. Li[1], Mingchuan Luo [1], Dae-Hyun Nam[1], Ning Wang[1], Tao Peng [1], Yu Yan[1], Geonhui Lee [1] & Edward H. Sargent [1✉]

Nitrogen-doped graphene-supported single atoms convert $CO_2$ to CO, but fail to provide further hydrogenation to methane – a finding attributable to the weak adsorption of CO intermediates. To regulate the adsorption energy, here we investigate the metal-supported single atoms to enable $CO_2$ hydrogenation. We find a copper-supported iron-single-atom catalyst producing a high-rate methane. Density functional theory calculations and in-situ Raman spectroscopy show that the iron atoms attract surrounding intermediates and carry out hydrogenation to generate methane. The catalyst is realized by assembling iron phthalocyanine on the copper surface, followed by in-situ formation of single iron atoms during electrocatalysis, identified using operando X-ray absorption spectroscopy. The copper-supported iron-single-atom catalyst exhibits a $CO_2$-to-methane Faradaic efficiency of 64% and a partial current density of 128 mA cm$^{-2}$, while the nitrogen-doped graphene-supported one produces only CO. The activity is 32 times higher than a pristine copper under the same conditions of electrolyte and bias.

---

[1] Department of Electrical and Computer Engineering, University of Toronto, Toronto, ON, Canada. [2] Department of Applied Chemistry, National Yang Ming Chiao Tung University, Hsinchu 300, Taiwan. [3] Taiwan Semiconductor Research Institute, National Applied Research Laboratories, Hsinchu 300, Taiwan. [4] These authors contributed equally: Sung-Fu Hung, Aoni Xu, Xue Wang, Fengwang Li. ✉email: ted.sargent@utoronto.ca

Metal nanostructures supported on host substrates, such as metal oxide, metal, and graphene, are widely used in heterogeneous catalysis[1,2]. The size of metal nanoclusters is a prime determinant of the catalytic characteristics of the resultant composite materials.

Recently, the ultimate small-size limit for metal catalysts—nitrogen-doped graphene-supported single atoms—has attracted interest for its potential in reducing $O_2$ and $CO_2$[3–7]. Isolated metal atoms immobilized on nitrogen-doped carbon substrates through metal-nitrogen coordination have shown excellent $O_2$-reducing activity, comparable to that of platinum catalysts[8,9], and have also enabled $CO_2$ reduction reaction (CO2RR) instead of the competing hydrogen evolution that is seen in the bulk forms of the same materials[10,11].

Until now, however, $CO_2$RR products from nitrogen-doped graphene-supported single atoms have been limited to CO: the weak binding of *CO intermediates led to facile release of gaseous CO[12–14]. We posited here that it would be possible to modulate $CO_2$RR selectivity toward hydrocarbons if we could regulate the electronic structure of the single-atom site by altering dramatically the choice of host substrate. In prior studies of metal-supported single atoms, i.e., metals incorporating atomically-dispersed elements[11,15], DFT calculations have predicted that the binding and activation energies of reaction intermediates on metal-supported single atoms can be tuned to promote catalytic behavior[16,17].

We pursued the conversion of $CO_2$ to hydrocarbons—methane in the present work—offering a means to store renewable electricity in the form of chemical fuels[18–21]. Enhancing catalytic activity and improving selectivity remain key goals[22–24], and, among $CO_2$RR products, methane is of interest in light of its established infrastructure for storage, distribution, and utilization. Prior catalysts have exceeded 50% of the selectivity for methane, but at lower than practical current densities[25–27], which limits industrial application: indeed, such analyses suggest an initial target selectivity of 60% combined with current density >100 mA cm$^{-2}$ [28,29].

## Results and discussion

We began by investigating, using DFT calculations, whether $CO_2$ methanation is feasible on atomically-dispersed elements on Cu(111), the preferred orientation in polycrystalline copper exhibiting the lowest surface energy of all low-index facets of Cu with an fcc crystal structure (Supplementary Table 1). We first examined, on the single-atom sites, the adsorption energies of *CO vs. *H and the hydrogenation energy of *CO (Fig. 1a). The adsorption of CO favors $CO_2$RR, while *H determines the ease of further reduction to methane[21,22,24]. We found that Fe single atoms, among a range of transition metals explored, exhibit the strongest *CO affinity over competing *H, and the lowest hydrogenation energy of *CO, suggesting a tendency for $CO_2$RR to products beyond CO. Interestingly, transition elements such as Co and Ni, show favorable trends for $CO_2$RR as well. The selectivity toward methane is higher for Cu-NiSA and Cu-CoSA (Fig. 1b) compared to pristine Cu, but lower than that when Fe is used.

With the view that particle size can influence catalytic activity—prior reports highlighted that aggregated forms of Fe prefer HER, while the single-atom form adopts $CO_2$RR[5,10]—we further examined the effect of Fe domain size, and found that the affinity of *CO and *H is dependent on the size of Fe in the Cu host. Fe atoms exhibit an increasing *CO affinity and a decreasing *H affinity as the Fe domain size decreases, and they reach the highest affinity for *CO over *H on single-site Fe (Fig. 1c). This size-dependent relationship is attributed to the fact that Fe single

atoms redistribute the d orbitals near the Fermi level compared to bulk counterparts (Supplementary Fig. 1), altering their adsorption properties and thus their catalytic selectivity. We compared the $CO_2$RR performance of Fe single-atom catalysts vs. aggregated Fe catalysts (Fig. 1d). Fe nanoparticles on the Cu surface produce hydrogen, consistent with prior reports[10]. Narrowing the size of Fe to clusters (Supplementary Fig. 2) produces a maximum methane FE of 39%. Cu-Fe clusters, accordingly, have exhibited a modest selectivity toward methane in a photoelectrochemical system[30], indicating the size effect on methane selectivity. Only by further reducing the Fe size to the single-atom level do we maximize selectivity toward methane.

In examining the catalyst's density of states by DFT calculations, we found that Fe single atoms offer d orbitals near the Fermi level (Fig. 2a): such electronic properties have previously been shown to facilitate the adsorption of reaction intermediates[31]. By deconvoluting these d orbitals, we found that they arise mainly from the z axis character (Fig. 2b and Supplementary Fig. 3), resulting in preferred adsorption of *CO on atop sites of Fe (Fig. 2c)[32]. In comparison, *CO binds at the hollow sites of Cu atoms (Fig. 2c and Supplementary Fig. 4). The binding of *CO to atop Fe is stronger than that to hollow Cu. In Fig. 2d, we find that *CO transfers from Cu nearby (bridge sites and atop sites) to atop Fe, indicated by arrowheads, while only Cu at hollow sites is stabilized at the original sites.

To study product selectivity, we calculated the post-adsorption steps on Cu-FeSA: CO hydrogenation and C-C coupling. Their energy level determines the branching of C1 (e.g., $CH_4$, $CH_3OH$) vs. C2 (e.g., $C_2H_4$, $C_2H_5OH$) products[21,22,24]. We found that C-C coupling is energetically unfavorable on Cu-FeSA (both on Cu-Cu and Cu-Fe sites) compared to on Cu (Fig. 2e), indicating that *CO adsorbed on atop Fe sites will not couple with neighboring *CO on Cu sites to generate C2 products. We further investigated the surface CO coverage effect on Cu-FeSA surface, shown in Supplementary Fig. 5. Hydrogenation and coupling reactions are promoted with increasing CO coverage, in agreement with previous studies[23,33,34], while $C_2$ will become favored at high coverage. Previous quantum mechanical studies predict that Au/Ag supported single atoms favor C1 products[17]. *CO undergoes hydrogenation to form *COH favorably over *CHO on the Fe sites of Cu-FeSA when there is a contribution of solvation (Fig. 2f and Supplementary Figs. 6, 7), while *CHO is preferred on pristine Cu (Supplementary Fig. 8). The overall $CO_2$-to-methane energy diagram for Cu-FeSA indicates that the rate determining step is *CO → *COH ($\Delta E = 0.52$ eV, Fig. 2g). Taken together, the DFT calculations suggest that Cu-FeSA favors the production of $CH_4$ via a pathway involving the hydrogenation of *CO to *COH instead of C-C coupling.

We pursued therefore the experimental synthesis of the metal-supported single atoms, a challenge in materials science due to the ease of aggregation of the single sites. We employed a planar complex, iron-phthalocyanine (FePc), as the precursor of the single-atom metal: this, we posited, could separate central Fe ions and minimize the distance between the Fe ion and the Cu surface. Because FePc does not interact with the Cu surface, FePc is not directly adsorbed on the Cu surface (Supplementary Fig. 9); we therefore functionalized the Cu surface with the aid of 3-mercaptopropionic acid (3-MPA), enabling the thiol end to anchor the Cu surface and the oxyl end to bond with the central Fe ions in FePc, forming Cu-FePc (inset of Fig. 3a). This approach was designed to avoid FePc stacks that could produce aggregation of Fe, and sought to ensure that FePc deposited on the Cu surface. Evidence of this is seen in depth-profiling X-ray photoelectron spectroscopy in Supplementary Fig. 10.

Using X-ray diffraction (XRD), we estimate distance between the Fe ion and Cu to be 7.8 Å (diffraction peak at 11.3° in Fig. 3a),

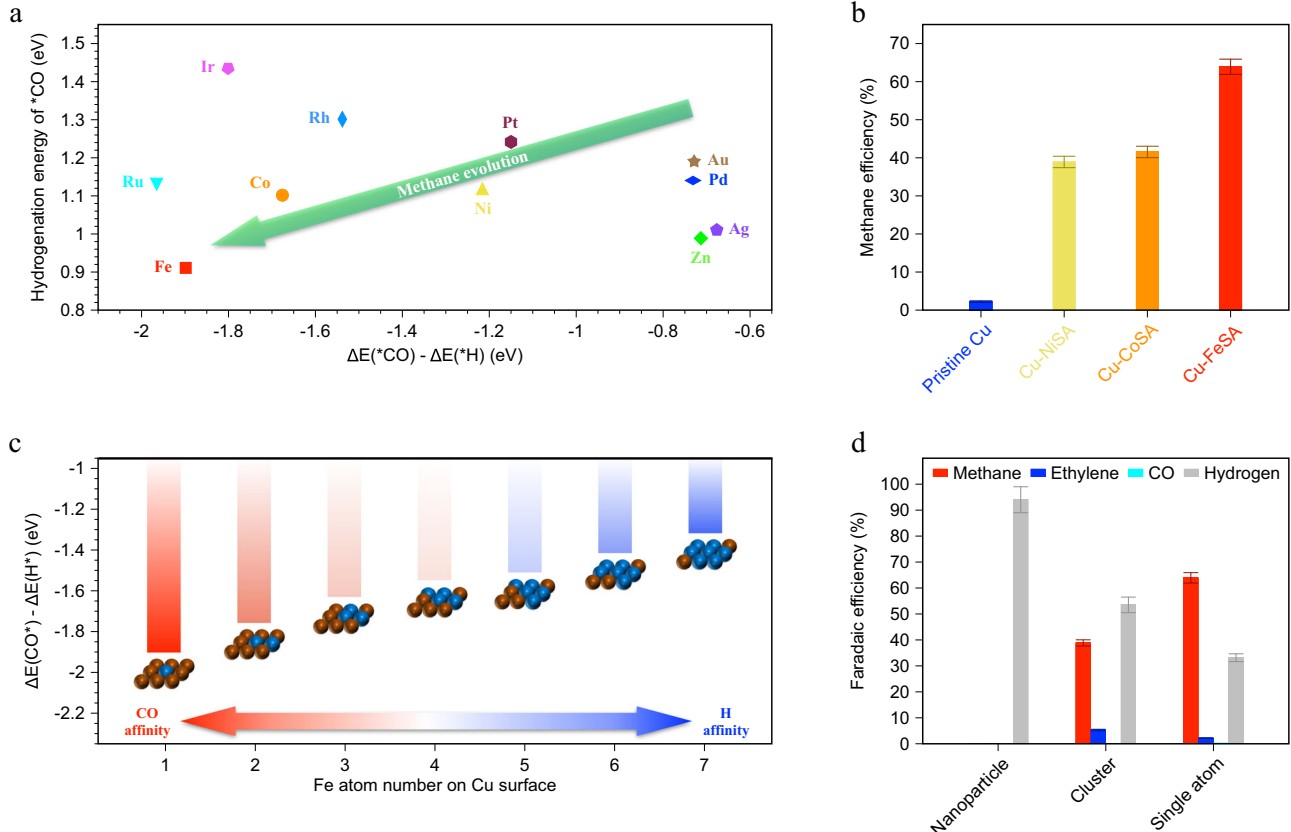

**Fig. 1 Calculations and catalytic activities for single-atom-anchored copper. a** Comparison of adsorption energy and hydrogenation of *CO on various single-atom catalytic sites. **b** Catalytic methane activities of pristine Cu vs. various single-atom-anchored copper catalysts for $CO_2$ reduction reaction. **c** Effect of iron domain sizes in the copper surface on the adsorption energies of *H and *CO. **d** Catalytic activities of various iron-dispersed copper materials, including nanoparticle, cluster, and single-atom forms. The error bars represent 1 s.d. on the basis of three independent samples.

matching the predicted distance Fe···3-MPA···Cu. We do not observe diffraction signals related to its stacking in FePc powder at 10°, 15.7°, and 16.3°[35]. We associate the grazing-incidence-angle XRD in Supplementary Fig. 11 (signal at 11.3°) with contributions from surface self-assembling FePc. The rough surface of sputtered copper provides ample adsorption sites for iron phthalocyanine, exhibiting the intensified monolayer XRD signal. We also conduct atomic force microscope studies; the roughness of ~9 nm on the sputtered copper (Supplementary Fig. 12), while a 9 Angstrom step height, corresponding to the predicted distance Fe···3-MPA···Cu, is observed on the iron-phthalocyanine-modified copper (Supplementary Fig. 13). The Cu surface becomes smoother following molecular assembly (Supplementary Fig. 14) but keeps a (111) preferred orientation, shown in Fig. 3a (the peak at 43.5°). Using X-ray absorption spectroscopy (XAS), we found that the intramolecular coordination environment of the oxyl-bonded FePc remains similar to that of unbound FePc (Fig. 3b), while the oxidation state of the central Fe increases slightly, a finding we ascribe to the additional coordination of the oxyl bond (Supplementary Fig. 15). We observe additional vacancies in the 3d orbitals of the central Fe atom (Supplementary Fig. 16), which we assign to bonding with the high electron-affinity oxyl end in 3-MPA. Focusing on the bonding environment of surface Cu (Supplementary Fig. 17), we observe an energy shift to the lower binding energy of the Cu peak in Cu-FePc, allowing us to identify Cu-S functional bonding during 3-MPA surface modification[36,37]. The shoulder at higher binding energies is attributed to the Cu-O due to exposure to air prior to XAS measurement. Overall, the XRD and XAS studies suggest the

formation of self-assembling FePc on Cu, without measurable evidence of stacking beyond a monolayer.

We applied a negative bias to enable the Fe-N cleavage and formation of Cu-FeSA. This provides in-situ synthesis during $CO_2RR$: the FePc reductive demetallation process initiates at potentials as positive as +0.4 V vs RHE[38], and continues in the potential range (−0.7 to −1.3 V vs RHE) of $CO_2RR$. We found, using in-situ XAS (Fig. 3c), that the Fe-N bond disappears and the Fe-Cu metallic bond (a lower coordination intensity compared to pure Fe metal) emerges during $CO_2RR$[39]. Accordingly, the oxidation state also changes from that of a cation to a metallic one, and the electronic state differs from that of pure Fe metal (Fig. 3d and Supplementary Figs. 18–20). The large diameter of the phthalocyanine ring (15 Å) separates iron ions (2.52 Å) well[40], preventing self-aggregation of Fe ions. We further confirmed the Cu-Fe bond from structural fitting of the extended X-ray absorption fine structure (EXAFS, Supplementary Fig. 21). We also found, from N 1s XPS (Supplementary Fig. 22), that the peak from N-Fe and N-C in FePc drops following electroreduction[41,42], suggesting the detachment of the phthalocyanine ring from the surface. We also investigated Cu K-edge XAS from which we found that Cu keeps its metallic state during $CO_2RR$ (Supplementary Figs. 23, 24). Using atomic-resolution transmission electron microscopy (Fig. 3e), we observe the dispersion of single Fe atoms in the Cu matrix. Single Fe atoms appear as bright spots, the result of their causing a more scattered electron beam received by the high-angle annular detector.

The distribution of Fe atoms in the as-formed Cu-FeSA is uniform, and we did not see evidence of an aggregated form of

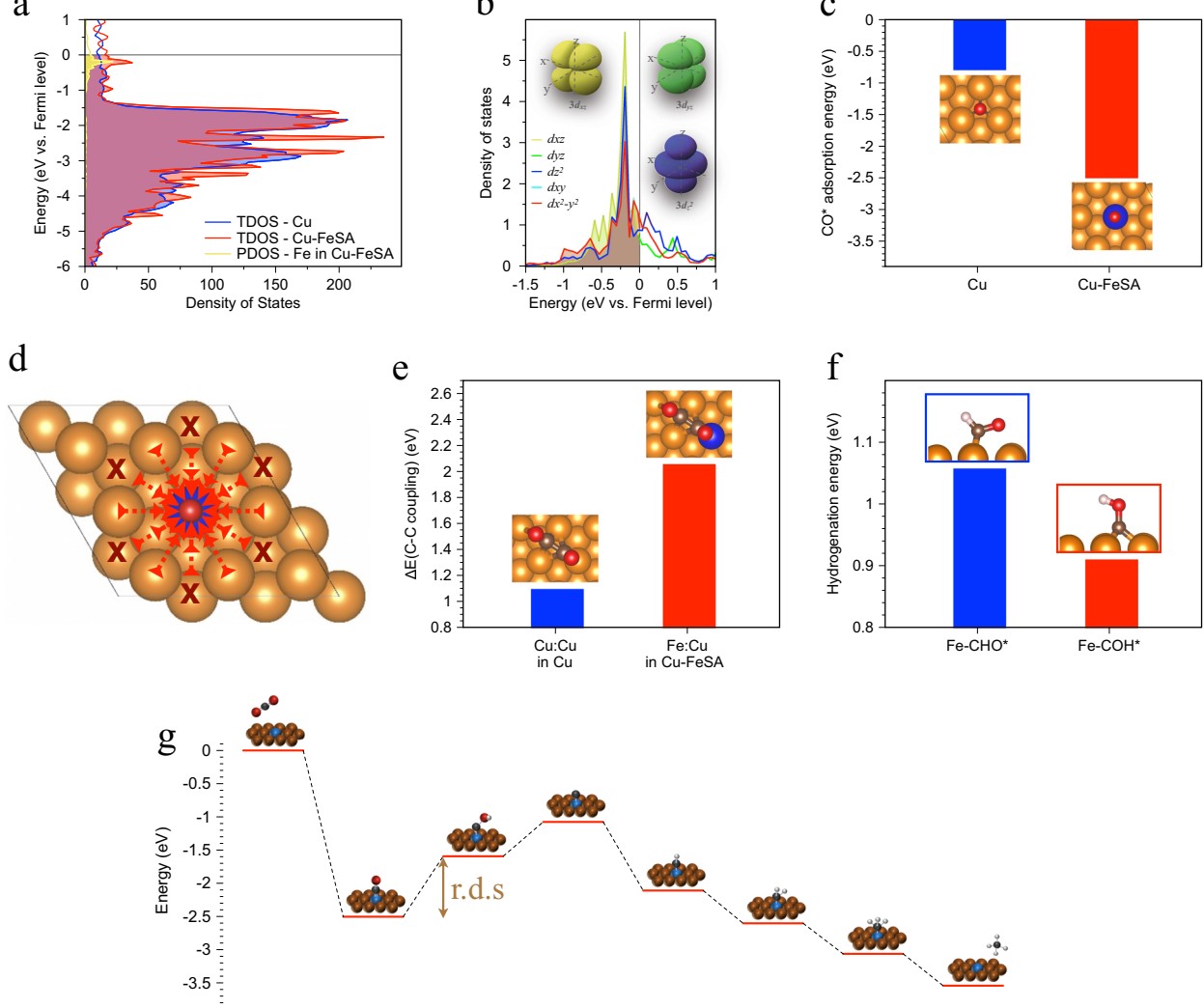

**Fig. 2 Mechanism study for the copper-supported single-atom iron catalyst. a** Density of states of pristine Cu and Cu-FeSA. **b** Adsorption energy of *CO for pristine Cu and Cu-FeSA. **c** Deconvolution of d orbitals of single-atom iron in Cu-FeSA. **d** Schematic illustration of *CO transition: the arrow heads mean the transition pathway while the cross marks suggest immobile *CO adsorption sites. **e** C-C coupling energy for pristine Cu vs. Cu-FeSA. **f** Hydrogenation energy of the intermediates for methane production on the iron sites in Cu-FeSA. **g** Energy diagram for methane evolution in Cu-FeSA. The r.d.s. (rate determining step) is the hydrogenation of *CO intermediates on the iron sites.

iron was found (SEM, TEM and elemental scans over scales from nanometers to micrometers, Supplementary Figs. 25–29). The Fe proportion obtained using inductively coupled plasma mass spectrometry (ICP-MS) is 0.3 at%.

In-situ Raman spectroscopy allows us to monitor materials synthesis and to identify reaction intermediates during $CO_2RR$. We first study the Raman spectra of bare Cu (Fig. 3f) as a reference. The initial faint peak at 520 cm$^{-1}$ is ascribed to copper oxides, a mixture of $Cu_2O$ and CuO via Cu L-edge soft XAS (Supplementary Fig. 30), consistent with the XPS results in Supplementary Fig. 16. During $CO_2RR$, the surface oxides are reduced and CO intermediates are adsorbed on Cu (*$CO_{Cu}$, 365 cm$^{-1}$ for Cu-CO stretch and 1079 cm$^{-1}$ for $CO_3^{2-}$ symmetric stretch on Cu surface)[43–46]. We also observe the intramolecular C≡O stretch on the Cu surface (Supplementary Fig. 31).

In examining the in-situ Raman spectra of the precursor film Cu-FePc, we identified the peaks at 660 cm$^{-1}$, 745 cm$^{-1}$, and 935 cm$^{-1}$ as the phthalocyanine vibrations in FePc, and the peak at 535 cm$^{-1}$ as the *CO adsorbed on iron sites (*$CO_{Fe}$, Fe-CO stretch)[38,47–49]. The Raman peaks of the phthalocyanine ring fade as the potential is decreased to −1.1 V vs. RHE and further

negative potentials, suggesting that the phthalocyanine ring has detached from the Cu surface, consistent with in-situ XAS results. The Fe-CO stretch remains after the removal of the phthalocyanine ring, confirming that the Fe atom is retained on the Cu surface. The intensity of *$CO_{Fe}$ is higher than that of *$CO_{Cu}$ in Cu-FeSA, agreeing with the view that Fe sites attract and convert $CO_2$ to *CO on Fe sites, and suppress adsorbed *CO on Cu sites. In the case of Cu-FeSA, we observe a small intramolecular C≡O stretch on the Cu surface under a low potential (Supplementary Fig. 31). This then vanishes under a high potential, and a new intramolecular C≡O stretch on the Fe site appears. This observation is in agreement with the suppression of *CO on Cu sites. We associate the 740 cm$^{-1}$ signal at −0.6 V vs RHE with 3-MPA (Supplementary Fig. 32) since it is closed to the Cu surface. At −0.7 ∼ −1.0 V vs RHE, the noticeable red spots can be attributed to FePc, which moves toward the copper surface and increases the Raman intensity as the 3-MPA detaches from the Cu surface. The intensity at 660 cm$^{-1}$ and 935 cm$^{-1}$ increases from −0.6 V to −0.7 V vs RHE and then decreases at a more negative voltage, suggesting that the phthalocyanine ring of FePc is detached from the Cu surface. We observe a signal associated with the

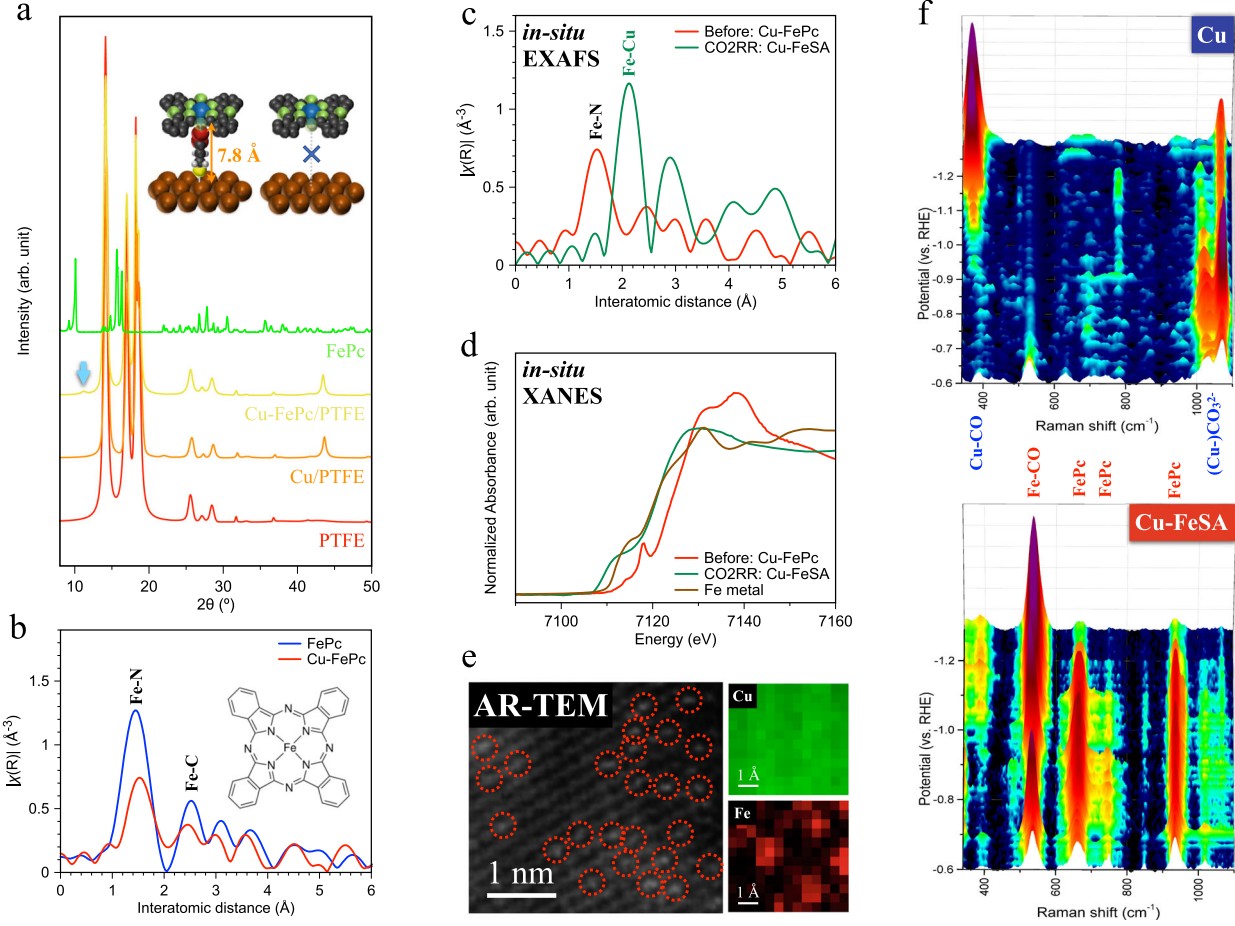

**Fig. 3 Materials characterization and in-situ investigation of iron-phthalocyanine-modified and iron-single-atom-anchored copper. a** X-ray diffraction. The inset illustrates the bonding between Cu surface and iron phthalocyanine using 3-mercaptopropionic acid. **b** Extended X-ray absorption fine structure (EXAFS) of Fe K-edge for the Cu-FePc GDE. **c** In-situ EXAFS and (**d**) in-situ XANES of Fe K-edge for identifying Cu-FeSA during $CO_2RR$. **e** Atomic resolution transmission electron microscope images and atomic elemental mapping using EELS. Dashed circles indicate the single-atom iron. **f** In-situ Raman spectroscopy for pristine Cu and Cu-FeSA. The intensity scale is 4000 c.p.s. in the spectrum.

phthalocyanine ligand that remains following $CO_2RR$, consistent with the view that iron atoms are attracted towards the copper surface and that some phthalocyanine rings remain near the copper surface, but they depart from the catalytic surface after approximately one hour. (Supplementary Fig. 33) Overall, we conclude that the Fe-N bond breaks during electrocatalysis, the phthalocyanine ring detaches from the central Fe ions, converting to Cu-FeSA. Cu-FeSA shows strong adsorption of *CO on Fe sites following the reduction reaction.

We investigated $CO_2RR$ in neutral electrolyte (1 M $KHCO_3$) in a flow cell. The neutral condition circumvents the consumption of $CO_2$ by alkaline electrolye[50], rendering it more compatible with sustained $CO_2RR$. The major product on the copper-supported iron-single-atom catalyst is methane, and its Faradaic efficiency (FE) reaches a maximum value of 64% at a current density of 200 mA cm$^{-2}$ (Fig. 4a), showing progress in the direction of technoeconomic promise (Supplementary Table 2), while the nitrogen-doped-graphene-supported one generates only carbon monoxide (Supplementary Fig. 34). In comparison, the major product on bare Cu is ethylene (maximum 54% FE at a current density of 200 mA cm$^{-2}$) and the methane FE is only 2% FE. The local pH in $KHCO_3$ at 200 mA cm$^{-2}$ is 12[51]. At high pH, C-C coupling dominates, and the C1 pathways are suppressed, facilitating selectivity to C2 products, in

line with the catalytic result seen on bare Cu[52]. Cu-FeSA shows a 32x FE to methane compared to bare Cu. The SAA system also gives a 1.3x higher FE to methane compared to the prior report[53]. We optimized catalytic performance by controlling several parameters: thickness of host copper layer, 3-MPA modification time, and FePc adsorption time (Supplementary Figs. 35–37). With optimal parameters, we achieved a methane partial current density of 128 mA cm$^{-2}$ with a turnover frequency (TOF) of 31770 h$^{-1}$, higher than 4.19 h$^{-1}$ for bare Cu (Fig. 4b, Supplementary Note 1). The TOF of Cu-FeSA for producing methane is higher than that of single-atom catalysts on nitrogen-doped carbon substrates for producing carbon monoxide (14800 h$^{-1}$)[3]. When we added potassium thiocyanate into the catholyte to poison Fe[54,55], methane selectivity dropped within several minutes: this suggests that the active site for the hydrogenation is Fe. Electrochemical impedance spectroscopy for Cu-FeSA (Supplementary Fig. 38) reveals a low $R_{CT}$: the surface is therefore highly activated in the range of potential we apply[56,57]. We detect a small amount of ethylene (2%) in Cu-FeSA (Supplementary Note 1 and Supplementary Figs. 39, 40).

The Cu-FeSA is stable during an electrolysis of 12 h: the methane FE and applied potential remain stable at a constant operating current density of 200 mA cm$^{-2}$ (Fig. 4c). We offer that

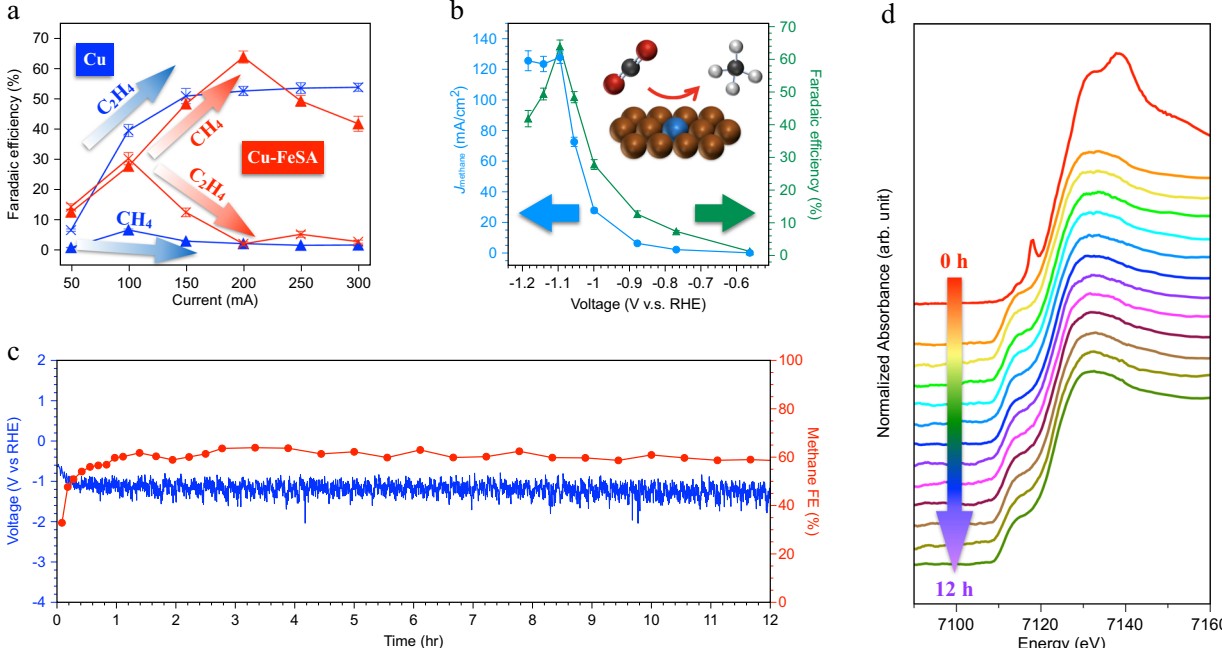

**Fig. 4 Catalytic performance of Cu-FeSA. a** Comparison of reaction products for pristine Cu and Cu-FeSA. Error bars represent 1 standard deviation on the basis of three independent samples. **b** Faradaic efficiency and partial current density to methane vs. applied potential. **c** Stability of methane production. **d** In-situ X-ray absorption near-edge structure (XANES) of Fe K-edge for long-term investigation over 12 h. The error bars represent 1 s.d. on the basis of three independent samples.

the activation process correlates with the fact that the phthalocyanine rings remain and then leave the catalytic surface, shown in Supplementary Fig. 33. We performed in-situ XAS to investigate the long-term evolution of electronic configurations of Fe (Fig. 4d). In-situ XANES provides oxidation number via analysis of the edge shift, and it provides the chemical orbital hybridization via the oscillation behavior near the white-light region. We did not observe an edge shift nor shape change near the white-light region. This suggests that the chemical environment of Cu-FeSA remains substantially unchanged during electromethanation. The Fe compositions, examined by ICP-MS, also remain similar before and after reaction. According to literature[58,59], Cu undergoes dynamic phase transformation/surface reconstruction during $CO_2RR$, affecting catalytic stability. We performed in-situ time-resolved X-ray absorption spectroscopy (XAS) to investigate the relevant phenomenon, shown in Supplementary Fig. 41. The sputtered Cu in Cu-FeSA exhibits a metallic state in 2 min and retains this over 30 min. We did not observe dynamic reoxidation/reduction behavior and phase transformation reported previously in H-cell systems. In-situ Raman spectroscopy in Fig. 3f showed that the peak for surface copper oxide continuously decreases with increasing applied potential, without reoxidation. We conclude that, in the flow-cell system, the catalyst undergoes the intensive driving force of reduction at 200 mA cm$^{-2}$, leading to undetectable reoxidation/reduction.

We developed Cu-based catalysts containing Fe single atoms for the electrochemical methanation of $CO_2$. We assembled Fe phthalocyanine onto the Cu surface and reduced it to Fe in a copper host during electrocatalysis. Fe attracts CO intermediates and contributes to their conversion through a COH intermediate to methane. We report $CO_2$-to-methane conversion having a Faradaic efficiency of 64% and a partial current density of 128 mA cm$^{-2}$, 32 times higher than Cu under the same conditions of electrolyte and bias. The more active single-atom Fe is present at the surface of the Cu and remains stable over the operating times considered herein.

## Methods

**Preparation of gas diffusion electrodes (GDEs).** All chemicals were purchased from Sigma–Aldrich and used without further purification. GDE was made via deposition of sputtered Cu on a PTFE substrate, followed by assembly of iron phthalocyanine (FePc) on the Cu surface. Cu with a thickness of 200 nm was sputtered on a PTFE substrate (pore size of 450 nm) using a pure Cu target (99.99%) with an Ar flow rate of 6 mtorr at an applied power of 85 Watts. Cu/PTFE was immersed in 10 mM 3-mercaptopropionic acid (3-MPA$_{(aq)}$) for 10 min to modify the functional group on the copper surface, followed by a deionized water rinse and dried with nitrogen. Surface modified copper was submerged in 1 mM FePc$_{(MeOH)}$ for 90 min, rinsed completely with methanol, and dried with nitrogen. The surface-modified substrate was soaked into CoPc$_{(MeOH)}$ and NiPc$_{(MeOH)}$ to obtain the Cu-CoPc and Cu-NiPc. For Iron nanoparticles and clusters, 8.8 mg of iron(II) chloride and 20 mg of poly(methacrylic acid) ligand were added into a screw-neck glass bottle containing 5 ml of methanol[60]. After the solution becoming uniform by sonication, 0.5 ml of freshly prepared NaBH$_4$ solution (5 mg ml$^{-1}$ in methanol) was injected into the solution under vigorous stirring (2000 r.p.m.). The stirring speed of the solution was kept the same for another 10 min. In the washing process, the precipitate was sonicated to ensure all clusters dispersing uniformly in methanol, and then separated by centrifuge.

**Materials characterization.** Microstructure and elemental analysis were collected by Hitachi SU5000 FE-SEM equipped Bruker energy dispersive X-ray spectroscopy. Cold-field emission Cs-corrected transmission electron microscope (JEOL ARM-200F) with 200 keV acceleration voltage was used in microstructure analysis at Material and Chemical Laboratories, Industrial Technology Research Institute (ITRI), Taiwan. Electron Energy Loss Spectrum (EELS) observations were performed using Gatan Imaging Filter (GIF) model 965 QuantumER™ at room temperature with camera length of 2 cm. The energy resolution measured by the full width at half maximum of a zero-loss peak was about 0.4 eV through a cold field emission gun. Surface morphology and step profile were measured using Veeco Dimension 5000 Scanning Probe Microscope (D5000) at Taiwan Semiconductor Research Institute (TSRI), Taiwan. Crystal structures were determined by X-ray diffraction (XRD) patterns, collected by Rigaku MiniFlex600 G6. The grazing-incidence-angle XRD patterns were acquired by PANalytical X'Pert Pro at Taiwan Semiconductor Research Institute (TSRI), Taiwan. X-ray photoelectron spectroscopy (XPS) measurements were conducted on a Thermo Scientific K-Alpha spectrophotometer with a monochromated Al Kα X-ray beam (1486.60 eV). The depth-profiling XPS spectrum were collected on the same model at Taiwan Semiconductor Research Institute (TSRI), Taiwan. All binding energies of the elements were calibrated to adventitious carbon at 284.50 eV. X-ray absorption spectroscopy (XAS) including X-ray absorption near edge spectra (XANES) and extended X-ray absorption fine structure (EXAFS) of Fe K-edge were collected by a silicon drift detector at ambient air at the 9BM beamline of the Advanced Photon Source (APS) located at Argonne National Laboratory and at the SXRMB beamline

of the Canadian Light Source (CLS). The pre-edge baseline was subtracted and the spectra was normalized to the post-edge. EXAFS analysis was conducted using a Fourier transform on $k^2$-weighted EXAFS oscillations to evaluate the contribution of each bond pair to the Fourier transform peak. REX2000 software using ab initio-calculated phases and amplitudes from the program FEFF 8.2 was used for EXAFS fitting. CN, R, $\Delta E$, and the EXAFS Debye–Waller factor (DW; $\sigma^2$) are variable parameters of the EXAFS equation for fitting the experimental result. XANES of Cu L-edge and Fe L-edge were obtained in total electron yield mode (TEY) and total fluorescence yield mode (TFY) at the SGM beamline of the Canadian Light Source (CLS). The iron concentration was quantified by inductively coupled plasma mass spectrometry (ICP-MS).

Electrocatalytic characterization. Electrochemical properties were investigated using an Autolab PGSTAT204 potentiostat in a flow cell reactor using a gas diffusion electrode (GDE) as the working electrode (WE), nickel foam as the counter electrode (CE) and a saturated Ag/AgCl electrode as the reference electrode (RE). The WE and CE were separated by an anion exchange membrane. Aqueous solution of 1.0 M KHCO$_3$ saturated with CO$_2$ (pH = 7.0) was used as the electrolyte and the flow rate of CO$_2$ was 50 sccm. The potentials were converted to values relative to RHE based on the following equation: $E_{RHE} = E_{Ag/AgCl} + 0.0591 \times pH + E^0_{Ag/AgCl}$, where $E^0_{Ag/AgCl}$ is the standard potential of Ag/AgCl relative to SHE at 25 °C (0.210 V). The working area of electrodes was 1.0 cm$^2$ for each experiment. Linear sweep voltammetry was performed at the scan rate of 20 mV/s with iR-correction. Rs and Nyquist plot were collected by electrochemical impedance spectroscopy (EIS) measurements with a frequency range from 0.1 to 10$^5$ Hz with AC amplitude of 10 mV. Cyclic voltammetry was measured at the scan rate of 100 mV/s. The gaseous products were evaluated via gas chromatography (PerkinElmer Clarus 600) equipped with a thermal conductivity detector and a flame ionization detector. TOFs (h$^{-1}$) for the CO$_2$RR were evaluated based on the 8-electron pathway for methane. TOF = Turnover number for methane formation/Number of active sites. The iron poisoning experiment was conducted by adding 0.01 M potassium thiocyanate into the catholyte after the single-atom iron formed during CO$_2$RR.

In-situ identification. We executed in-situ X-ray absorption spectroscopy in the same conditions as electrochemical testing using a customized flow cell with an opening sealed by Kepton tape in the gas chamber. We collected the XAS signal using a Vortex detector at the 9BM beamline of the Advanced Photon Source (APS). We conducted in-situ time-resolved XAS at the 44A beamline of the Taiwan Photon Source (TPS, NSRRC, Taiwan). In-situ Raman measurements were conducted using a Renishaw inVia Raman microscope in a modified flow cell with a water immersion objective, employing a diode laser at 785 nm. CO$_2$ with a flow rate of 40 sccm flowed through the gas chamber. An Ag/AgCl (3 M KCl) electrode and a Pt wire were used as the reference and counter electrodes, respectively, for the in-situ Raman measurements.

Theoretical calculations. We performed Density functional theory calculations with the Vienna Ab Initio Simulation Package (VASP) code[61,62]. The exchange correlation energy was modeled by using Perdew-Burke-Ernzerhof (PBE) functional within the generalized gradient approximation (GGA)[63]. The projector augmented wave (PAW) pseudo-potentials[64] were used to describe ionic cores. The cutoff energy of 450 eV was adopted after a series of tests. A Methfessel-Paxton smearing of 0.05 eV to the orbital occupation is applied during the geometry optimization and for the total energy computations. In all calculations, the atoms at all positions have Hellmann–Feynman forces lower than 0.02 eV Å$^{-1}$ and the electronic iterations convergence was 10$^{-5}$ eV using the Normal algorithm. A 6-layer (4 × 4) Cu (111) supercell was built to simulate the exposed surface of copper accompanying with a sufficient vacuum gap of 15 Å. Iron atom was exposed on the surface by replacing a Cu atom. Structural optimizations were performed on all modified slab models with a grid of (3 × 3 × 1) k-point. During the adsorption calculations, the top three layers are fully relaxed while the other layers are fixed at the tested lattice positions. The detailed model was referred to Supplementary Note 2.

## Data availability

The data supporting this study are available within the paper and the Supplementary Information. All other relevant source data are available from the corresponding authors upon reasonable request.

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

## Acknowledgements
The authors acknowledge funding support from Natural Gas Innovation Fund, the Natural Sciences and Engineering Research Council (NSERC) of Canada, and Natural Resources Canada Clean Growth Program, and Ontario Research Fund – Research Excellence program. All DFT computations were performed on the Niagara super-computer at the SciNet HPC Consortium. SciNet is funded by the Canada Foundation for Innovation, the Government of Ontario, Ontario Research Fund Research Excellence Program, and the University of Toronto. The authors thank T. Wu and G. Sterbinsky for technical support at the 9BM beamline of the Advanced Photon Source (Lemont, IL). This research used resources of the Advanced Photon Source, an Office of Science User Facility operated for the US Department of Energy (DOE) Office of Science by Argonne National Laboratory and was supported by the US DOE under Contract no. DE-AC02-06CH11357, and the Canadian Light Source and its funding partners. The authors thank Y. Hu, Q. Xiao, M. Shakouri at the 06B1-1 (SXRMB) beamline, and T. Regier and J. Dynes at the 11ID-1 (SGM) at the Canadian Light Source for their technical support. The authors also thank the help from Material and Chemical Laboratories, Industrial Technology Research Institute (ITRI), Taiwan, for STEM/EELS observation and FIB sample preparation. Dr. Mu Tung Chang and Dr. Ren-Fong Cai are responsible for acquiring HADDF STEM atomic image as well as the EELS spectrum analysis. Dr. Shih-Yi Liu and Ms. Mei Lun Wu are responsible to prepare plane-view TEM sample and sample pretreatment before FIB. S.-F.H. acknowledges support from Postdoctoral Research Abroad Program, Ministry of Science and Technology, Taiwan (Contract No. MOST 108-2917-I-564-016, MOST 110-2113-M-009-007-MY2 and MOST 110-2628-M-A49-002) and from the Yushan Young Scholar Program, Ministry of Education, Taiwan.

## Author contributions
E.H.S. supervised the project. S-F.H. conceived the idea and carried out the experiments. S.-F.H. and E.H.S. wrote the paper. A.X. carried out the DFT calculations. X.W. and F.L. conducted the in-situ Raman measurements. S-F.H. performed the in-situ XAS measurements. S.-H.H. conducted and analyzed the AFM, depth-profiling XPS, and grazing-incidence-angle XRD. Y.L. and E.G.C. carried out part of electrochemical experiments. A.S.R. conducted the XPS measurement. X.W., F.L., Y.C.L., M.L., J.W., D-H.N., and N.W. assisted in data analysis and manuscript writing. T.P., Y.Y., and G.L. helped to characterize the materials. All authors discussed the results and assisted during manuscript preparation.

## Competing interests
The authors declare no competing interests.
