## [Peer review file · Nature Communications]

REVIEWER COMMENTS

Reviewer #1 (Remarks to the Author):

This is a clever writing way to report the results. The copper-supported iron-single-atom catalyst, Cu-FeSA is essentially the same with the explosively reported copper based electrocatalysts. In the materials aspects, the Cu-FeSA was achieved by trapping iron phthalocyanine on copper surface and subsequent in situ reduction by applying electric potential. This, in other way, revealed the unstable characteristic of the metal phthalocyanine. I agree this would be a way to prepare single atom catalyst, however few advantages of this method can be expected. In the aspect of electroreduction performance, the CO₂-to-methane Faradaic efficiency of 64% at a partial current density of 128 mA cm⁻² in GDE electrode is reached. This is also not impressive. It is straightforward to understand the different performance as compared to nitrogen-doped graphene-supported Fe as well as the 32 times enhanced activity than pristine copper. I find no major flaws in experiments and data analysis, however, there is lack of novelty for this journal. Thus, I cannot recommend its publication.

Reviewer #2 (Remarks to the Author):

The electrochemical hydrogenation of CO₂ to methane (CH₄) represents a promising mean to store renewable electricity in the form of chemical fuels. In this work, the author reported that the Cu support Fe single atom is an excellent electrocatalyst toward CO₂ reduction to methane with a high Faradaic efficiency of 64% and a partial current density of 128 mA cm⁻². Through a series of experimental characterization combined with theoretical calculations, they explored the origin of the high activity of the catalyst. The results are impressive and this work is well-organized. However, several issues should be addressed before I can recommend publication.

1. It is known that Cu and Fe are vulnerable to the oxidation and leaching in harsh chemical environments. Several works reveal that in aqueous electrolyte Cu may spontaneously oxidize and alter the product profile of CO₂RR (J. Am. Chem. Soc. 2020, 142, 28, 12119–12132; ACS Cent. Sci. 2019, 5 (12), 1998–2009). Moreover, the nanoparticles with large surface area often suffer from the oxidation or aggregation problems. Therefore, it is necessary to emphasize the dynamic phase transformation and/or surface reconstruction during the cathodic CO₂RR process.
2. The selectivity of CO₂ reduction is of great importance. Previous studies have shown that local *CO coverage affects the distribution of products in CO₂RR (ACS Catal. 2017, 7, 1749; Nat. Catal. 2019, 2, 1124). Therefore, the relationship between local CO surface coverage and methane selectivity over Cu-supported Fe monatomic catalysts should be analyzed.
3. Previous studies have pointed out that the CH₄ generation of copper-based catalysts depends on the pH value. If the pH value of the electrolyte is high, the early CC substitution can dynamically inhibit the C1 pathway (Phys. Chem. Chem. Phys. Phys, 2015, 17, 17) 18924). J. Am. Chem. Soc. 2016, 138, 483). Therefore, the variation of the local pH near the electrode surface may affect the product distribution. The authors need to comment on this issue.
4. Supplementary Fig. 27 mentioned shorter 3-MPA modification time results in fewer single iron atoms, the figure lacks the corresponding catalytic activity, please add the data in this respect.

Reviewer #3 (Remarks to the Author):

Please refer to the attached review report.

Manuscript ID: NCOMMS-21-10097-T

“A Metal-supported Single-atom Catalytic Site Enables Carbon Dioxide Hydrogenation”

Reviewer #1 (Remarks to the Author):

This is a clever writing way to report the results. The copper-supported iron-single-atom catalyst, Cu-FeSA is essentially the same with the explosively reported copper based electrocatalysts. In the materials aspects, the Cu-FeSA was achieved by trapping iron phthalocyanine on copper surface and subsequent in situ reduction by applying electric potential. This, in other way, revealed the unstable characteristic of the metal phthalocyanine. I agree this would be a way to prepare single atom catalyst, however few advantages of this method can be expected. In the aspect of electroreduction performance, the CO₂-to-methane Faradaic efficiency of 64% at a partial current density of 128 mA cm⁻² in GDE electrode is reached. This is also not impressive. It is straightforward to understand the different performance as compared to nitrogen-doped graphene-supported Fe as well as the 32 times enhanced activity than pristine copper. I find no major flaws in experiments and data analysis, however, there is lack of novelty for this journal. Thus, I cannot recommend its publication.

Response: We have revised the manuscript to be clear that we achieved one of the best CO₂-to-methane Faradaic efficiency, equal to FE=64%. Given the referee's comment on nitrogen-doped graphene-supported Fe, we study this reference catalyst too, and obtain CO alone. The copper-supported iron-single-atom catalyst, Cu-FeSA, majorly produces CH₄ rather than CO. It shows that the metal-supported iron-single-atom catalyst is a new catalyst system, broadening the understanding of single-site catalysts in CO₂ reduction.

Reviewer #2 (Remarks to the Author):

The electrochemical hydrogenation of CO₂ to methane (CH₄) represents a promising mean to store renewable electricity in the form of chemical fuels. In this work, the author reported that the Cu support Fe single atom is an excellent electrocatalyst toward CO₂ reduction to methane with a high Faradaic efficiency of 64% and a partial current density of 128 mA cm⁻². Through a series of experimental characterization combined with theoretical calculations, they explored the origin of the high activity of the catalyst. The results are impressive and this work is well-organized. However, several issues should be addressed before I can recommend publication.

1. It is known that Cu and Fe are vulnerable to the oxidation and leaching in harsh chemical environments. Several works reveal that in aqueous electrolyte Cu may spontaneously oxidize and alter the product profile of CO₂RR (J. Am. Chem. Soc. 2020, 142, 28, 12119–12132; ACS Cent. Sci. 2019, 5 (12), 1998–2009). Moreover, the nanoparticles with large surface area often suffer from the oxidation or aggregation problems. Therefore, it is necessary to emphasize the dynamic phase transformation and/or surface reconstruction during the cathodic CO₂RR process.

Response: We now report in-situ time-resolved X-ray absorption spectroscopy (XAS) with the goal of investigating the dynamic reoxidation/reduction behavior and the phase transformation. This is now provided in Revised Supplementary Fig. S36. Time-resolved Cu K-edge XAS allows us to collect the spectrum every 2 min rather than every 30 min. We report that, at 200 mA cm⁻², sputtered Cu in Cu-FeSA exhibits a metallic state within 2 min, and remains in this state over continuous operation of 30 min. We better explain that we do not observe the dynamic reoxidation/reduction behavior and the phase transformation seen in prior reports based instead on H-cell systems. Using in-situ Raman spectroscopy (Fig. 3f), we report that the peak for surface copper oxide decreases progressively with increasing applied potential, and we do not see evidence of reoxidation. We now discuss our findings and explain that, in a flow-cell system, the catalyst undergoes a strong driving force of reduction at 200 mA cm⁻², leading to the lack of detected reoxidation/reduction.

These considerations inform this new main manuscript text:

“According to literature,^{58,59} Cu can undergo a dynamic phase transformation and/or surface reconstruction during CO₂RR. We performed in-situ time-resolved X-ray absorption spectroscopy (XAS) to investigate this possibility, shown in Supplementary Fig. 40. We found that the sputtered Cu in Cu-FeSA exhibits a metallic state in 2 min, and remains in this state following 30 minutes of continuous operation. We did not observe dynamic reoxidation/reduction behavior and phase transformation reported in prior H-cell system studies. In-situ Raman spectroscopy (Fig. 3f) indicated that the peak for surface copper oxide decreases progressively with increasing applied potential, without reoxidation. In flow-cell system, the catalyst undergoes an intensive driving force of reduction at 200 mA cm⁻², accounting for the lack of detectable reoxidation/reduction.”

“In-situ time-resolved XAS was conducted at the 44A beamline of the National Synchrotron Radiation Research Center (NSRRC, Hsinchu, Taiwan).”

Revised Supplementary Fig. 40 | In-situ time-resolved Cu K-edge X-ray absorption spectroscopy operating in flow cell at 200 mA cm⁻². (a) In-situ time-resolved X-ray absorption near-edge structure (XANES) for Cu K-edge. (b) In-situ time-resolved extended X-ray absorption fine structure (EXAFS) for Cu K-edge.

2. The selectivity of CO₂ reduction is of great importance. Previous studies have shown that local *CO coverage affects the distribution of products in CO₂RR (ACS Catal. 2017, 7, 1749; Nat. Catal. 2019, 2, 1124). Therefore, the relationship between local CO surface coverage and methane selectivity over Cu-supported Fe monatomic catalysts should be analyzed.

Response: We have now provided DFT calculations that seek to investigate the coverage effect in the calculation of the CO hydrogenation step (to methane) and coupling step (to C₂ products), shown below and in Supplementary Fig. 5. We find that both hydrogenation and coupling reactions are promoted with increased CO coverage, consistent with the previous studies, (ACS Catal. 2017, 7, 1749 and Nat. Catal. 2019, 2, 1124) but that the latter exhibits a more acute decline in reaction energies as a function of increasing coverage, agreeing with preferred coupling in the case of high CO coverage. We further compared energy difference between coupling step and hydrogenation step to identify the preference of CO coupling (Supplementary Fig. 5b). The reaction energy of CO coupling gradually decreases and becomes close to equaling the hydrogenation energy, indicating an improved priority of C₂ products at high coverage.

The full revised context is shown below and in Main text:

“We further investigated the surface CO coverage effect on Cu-FeSA surface, shown in Supplementary Fig. 5. Hydrogenation and coupling reactions are promoted with increasing CO

coverage, in agreement with previous studies,^{23,33,34} while an improved priority of C₂ products at high coverage.”

Supplementary Fig. 5 | Coverage effect. (a) reaction energy for CO hydrogenation (*COH and *CHO) and CO coupling steps on Cu-FeSA at different CO coverage; both hydrogenation and coupling reactions are promoted with increasing CO coverage (b) preference of CO coupling (defined as reaction energy difference between coupling step and hydrogenation step) at different CO coverage. With CO coverage increases from 2/9 (2 *CO on surface with 9 metal atoms) to 5/9, the reaction energy of CO coupling gradually decreases and close to hydrogenation energy, which suggests an improved priority of C₂ products at high coverage.

3. Previous studies have pointed out that the CH₄ generation of copper-based catalysts depends on the pH value. If the pH value of the electrolyte is high, the early CC substitution can dynamically inhibit the C1 pathway (Phys. Chem. Chem. Phys. Phys, 2015, 17, 17) 18924). J. Am. Chem. Soc. 2016, 138, 483). Therefore, the variation of the local pH near the electrode surface may affect the product distribution. The authors need to comment on this issue.

Response: We now better discuss the relationship to previous work (Nature **2020**, 557, 509, figure shown below): we discuss that, in that work, the local pH in KHCO₃ at 200 mA cm⁻² was 12.1. Thus, as seen in J. Am. Chem. Soc. 2016, 138, 483, at high pH, C-C coupling through adsorbed CO dimerization dominates, suppressing C1 pathways, thereby increasing selectivity to multi-carbon products. This matches with the catalytic result seen using pristine Cu, whose major product is ethylene (FE > 50%). Under the same catalytic conditions, Cu-FeSA catalyzes CO₂ into methane (C1 product). The Raman spectrum in Fig. 3f showed that Fe sites attract and convert CO₂ to *CO on Fe sites, and suppress adsorbed *CO on Cu sites. The more active single-atom Fe is present at the surface of the Cu and increases methane production, even at high pH.

The full revised context is shown below and in Main text:

“In comparison, the major product on bare Cu is ethylene (maximum 54% FE at a current density of 200 mA cm⁻²) and the methane FE is only 2% FE. The local pH in KHCO₃ at 200 mA cm⁻² is 12.⁵¹ At high pH, C-C coupling dominates, and the C1 pathways are suppressed, facilitating the selectivity of C2 products, in line with the catalytic result of bare Cu.⁵²”

Supplementary Fig. 12 in Nature 2020, 557, 509 | Local species modelling in the catalyst layer at different current densities. a, Concentrations of OH⁻, CO₃²⁻, HCO₃⁻ and CO₂. **b,** Local pH.

4. Supplementary Fig. 27 mentioned shorter 3-MPA modification time results in fewer single iron atoms, the figure lacks the corresponding catalytic activity, please add the data in this respect.

Response: We have added the data for the case of the shorter 3-MPA modification time. The catalytic activity showed less methane production than the optimized one (47% instead of 64%).

Revised Supplementary Fig. 35 | Catalytic activities of Cu-FeSA at 200 mA cm⁻² with various 3-MPA dip time. 3-MPA modification time affects the density of the anchored 3-MPA. A lower density

of the anchored 3-MPA due to the shorter 3-MPA modification time results in less anchored FePc and fewer single iron atoms. Otherwise, a high density of the anchored 3-MPA will interfere with the bonding process because one of the anchored 3-MPA bonds with the central Fe atom of FePc, but the other anchored 3-MPA nearby cannot bond with the phthalocyanine ring and prevent the access of FePc to the Cu surface.

Reviewer #3 (Remarks to the Author):

Hung et al. reported an interesting observation to selectively generating methane by CO₂ electrochemical reduction using metal-supported single atom catalysts. The observation serves as the common interest of the scientific community for the ultimate development of CO₂ conversion technology. His finding could be critical to the improvement of CO₂ electrochemical reduction; however, more elaborated scientific presentation should be provided before being accepted on Nat Communication. This manuscript is recommended for a major revision subject to further review, and the following comments need to be addressed in the revision.

Comments to general presentation:

1. In page 4, The authors claimed "we calculated the post-adsorption steps on Cu-FeSA: CO hydrogenation and C-C coupling, which determine branching of C1 (e.g. CH₄, CH₃OH) vs. C2 (e.g. C₂H₄, C₂H₅OH) products.^{21,22,24}". First of all, the wording is miss-leading since Ref 21, 22, 24 are published papers from other research groups, not the current work. This sentence should be substantially revised to avoid confusion.

Response: We have revised the sentence for accuracy to read:

"To estimate predicted product selectivity, we calculated the post-adsorption steps on Cu-FeSA: CO hydrogenation and C-C coupling. Their energy level determines the branching of C1 (e.g. CH₄, CH₃OH) vs. C2 (e.g. C₂H₄, C₂H₅OH) products.^{21,22,24}"

2. In page 4, the authors claimed "it is unlikely for *CO adsorbed on atop Fe sites to bond with neighboring *CO on Cu sites to generate C2 products, consistent with 17 prior theoretical studies. ". Since ref 17 was done on Au or Ag support, the so called "consistency" need to be re-elaborated.

Response: We have rewritten:

"We found that C-C coupling is energetically unfavorable on Cu-FeSA (both on Cu-Cu and Cu-Fe sites) compared to on Cu (Fig. 2e), indicating that *CO adsorbed on atop Fe sites is hard to bond with neighboring *CO on Cu sites to generate C2 products. We further investigated the surface CO coverage effect on Cu-FeSA surface, shown in Supplementary Fig. 5. Hydrogenation and coupling reactions are promoted with increasing CO coverage, in agreement with previous studies,^{23,33,34} while an improved priority of C₂ products at high coverage. The previous quantum mechanical study predicts that Au/Ag supported single atom favors the C1 products.¹⁷"

Comment to theoretical results:

1. As described in page 4 of supplementary information, the authors mentioned the use of ($\Delta G = \Delta E - T\Delta S$) for all of the calculations in this study. The definition of ΔG is not correct since

thermodynamics tells us $\Delta G = \Delta H - T\Delta S$. Thus, the definition of the calculated free energy should be comprehensively described in the context.

Response: We have rewritten Note 2 of Supplementary Information:

“The reaction energies for these were obtained by

$$\Delta G = \Delta E^{\text{DFT}} + \Delta \text{ZPE} - T\Delta S$$

where ΔE^{DFT} is the reaction energy calculated from DFT; ΔZPE is zero-point energy, which was neglected here; ΔS is the change in entropy, whose values of gaseous molecules are taken from the standard database in the NIST web-book.¹

Taking the CO hydrogenation reaction $*\text{CO} + \text{H}_2\text{O} + e^- \rightarrow *\text{COH} + \text{OH}^-$ as example, the reaction energy is equals to

$$\Delta G = G(*\text{COH}) - G(*\text{CO}) + G(\text{OH}^-) - G(\text{H}_2\text{O}) - G(e^-)$$

where, $G(*\text{COH})$ and $G(*\text{CO})$ is the total free energy of $*\text{COH}$ and $*\text{CO}$ adsorption configuration; $G(\text{OH}^-)$ is the free energy of hydroxyl ion, to calculate this value, we assume the equilibrium

Which relates the chemical potentials as

$$\mu_{\text{H}^+} + \mu_{\text{OH}^-} = \mu_{\text{H}_2\text{O}}$$

Then,

$$\mu_{\text{H}^+} + \mu_{e^-} + \mu_{\text{OH}^-} - \mu_{e^-} = \mu_{\text{H}_2\text{O}}$$

Thus,

$$\mu_{\text{OH}^-} - \mu_{e^-} - \mu_{\text{H}_2\text{O}} = -(\mu_{\text{H}^+} + \mu_{e^-})$$

Here, $\mu_{\text{H}^+} + \mu_{e^-}$ can be calculated using computational hydrogen electrode (CHE) model developed by Nørskov and co-workers.^{2,3}”

2. It is not clear that if the calculated free energy contains the contribution of solvation. From the context provided in the manuscript, there is no solvation contribution. The authors should justify the importance of solvation effect for describing HYDROGENATION processes in the simulation models.

Response: We now consider the contribution of solvation by using 6 H₂O molecules in the simulation model to describe the hydrogenation processes. We have calculated the hydrogenation energy for pure Cu and Cu-FeSA with H₂O in models and added these results into Supplementary Figs. S6 and S7. As shown in Supplementary Figs. S6 and S7, solvent decreases the hydrogenation energy on both Cu and Cu-FeSA in the pathways of $*\text{COH}$ and $*\text{CHO}$, but it does not change the catalytic mechanism and behavior. Through the investigation of the solvent effect, we conclude that Cu-FeSA still contributes to the $*\text{COH}$ pathway.

The full revised context is shown below and in Main text:

“*CO undergoes hydrogenation to form *COH favorably over *CHO on the Fe sites of Cu-FeSA whether there is a contribution of solvation (Fig. 2f and Supplementary Figs. 6 and 7), while *CHO is preferred on the pristine Cu (Supplementary Fig. 8).”

Supplementary Fig. 6 | Solvation effect on the CO hydrogenation through *COH pathway. (a) configuration of *COH adsorbed on Cu-FeSA with 6 H₂O molecules in model; (b) hydrogenation energy results of Cu and Cu-FeSA with and without solvation effect. Solvent decreases the hydrogenation energy on both Cu and Cu-FeSA in the pathways of *COH, but it does not change the catalytic mechanism and behavior.

Supplementary Fig. 7 | Solvation effect on the CO hydrogenation through *CHO pathway. (a) configuration of *CHO adsorbed on Cu-FeSA with 6 H₂O molecules in model; (b) hydrogenation energy results of Cu and Cu-FeSA with and without solvation effect. Solvent decreases the hydrogenation energy on both Cu and Cu-FeSA in the pathways of *CHO, but it does not change the catalytic mechanism and behavior.

3. Cu(111) surface was selected as the basal plane in the simulations. The rationale as well as the experimental evidence to support the choice of Cu(111) is not provided in the manuscript.

Response: We now discuss more clearly the calculated surface energy results seen in Supplementary Table S1. The (111) facets have the lowest surface energy among all low-index facets of Cu, with an fcc crystal structure, indicating that (111) is the most stable facet. For experimental evidence, the XRD pattern in Fig. 3a shows that the preferred orientation of the sputtered Cu is (111) facet before and after the molecular assembly. We now write:

“We began by investigating, using DFT calculations, whether CO₂ methanation is feasible on atomically-dispersed elements on Cu(111), the preferred orientation in polycrystalline copper exhibiting the lowest surface energy in all low-index facets of Cu with an fcc crystal structure (Supplementary Table 1).”

“The Cu surface becomes smoother following molecular assembly (Supplementary Fig. 14) but keeps (111) preferred orientation, shown in Fig. 3a (the peak at 43.5°).”

Supplementary Table 1. Surface energy results of low-index facets of Cu from DFT calculations

Cu facets	Surface energies (eV/ Å ²)
(100)	0.076
(110)	0.087
(111)	0.066

Fig. 3 | Materials characterization and *in-situ* investigation of iron-phthalocyanine-modified and iron-single-atom-anchored coppers. (a) X-ray diffraction. The inset figure illustrates the bonding between Cu surface and iron phthalocyanine using 3-mercaptopropionic acid. (b) Extended X-ray absorption fine structure (EXAFS) of Fe K-edge for the Cu-FePc GDE. (c) *In-situ* EXAFS and (d) *in-situ* XANES of Fe K-edge for identifying Cu-FeSA during CO₂RR. (e) Atomic resolution transmission electron microscope images and atomic elemental mapping using EELS. Dashed circles indicate the single-atom iron. (f) *In-situ* Raman spectroscopy for pristine Cu and Cu-FeSA.

4. Mechanism of C1 channel: the hydrogen of *CO to *COH (*CHO) on pristine Cu(111) was not shown using the current model design. How is it in comparison with Cu-FeSA model?

Response: We now add the CO hydrogenation energy in the case of pristine Cu as Supplementary Fig. 8 below. Pure Cu has a lower (~0.3 eV) hydrogenation energy through the *CHO reaction pathway; but the energy difference ~1 eV of CO coupling between pure Cu and CuFe (Figure 2e) suggests a higher preference for C2 products on pure Cu compared to Cu-FeSA. We now write:

“*CO undergoes hydrogenation to form *COH favorably over *CHO on the Fe sites of Cu-FeSA whether there is a contribution of solvation (Fig. 2f and Supplementary Figs. 6 and 7), while *CHO is preferred on the pristine Cu (Supplementary Fig. 8).”

Supplementary Fig. 8 | Hydrogenation energy of the intermediates for methane production on the pristine Cu. Even though pure Cu has a lower (~0.3 eV) hydrogenation energy through *CHO reaction pathway, an energy difference ~1 eV of CO coupling between pure Cu and CuFe suggests a higher preference for C2 products on pure Cu compared to CuFe SAA.

5. Mechanism of C2 channel: a common critical C-C bond coupling step, $*CO + *CO \rightarrow *OCCO$, was not taken into account in this study? Is any evidence to exclude this process?

Response: We have corrected *CO-to-*OCCOH to *CO-to-*OCCO, and replotted Figure 2e as below.

Fig. 2 | Mechanistic study of a copper-supported single-atom iron catalyst. (a) Density of states of pristine Cu and Cu-FeSA. (b) Adsorption energy of *CO for pristine Cu and Cu-FeSA. (c) Deconvolution of d orbitals of single-atom iron in Cu-FeSA. (d) Schematic illustration of *CO adsorption sites. (e) C-C coupling energy for pristine Cu vs. Cu-FeSA. (f) Hydrogenation energy of the intermediates for methane production on the iron sites in Cu-FeSA. (g) Energy diagram for methane evolution in Cu-FeSA. The r.d.s. (rate determining step) is the hydrogenation of *CO intermediates on the iron sites.

Comment to experimental results:

1. The authors employed AFM to show the dense and flat coating of iron-phthalocyanine on the copper surface. They should also display the AFM image of a pure copper surface and compare the images.

Response: We now better explain the atomic force microscope images of sputtered copper on flat Si wafer shown in Supplementary Fig. 12. The images display the surface of sputtered copper, whose roughness is approximately 9 nm. The rough surface provides abundant adsorption sites for iron phthalocyanine. The catalyst surface becomes smoother on the iron-phthalocyanine-modified copper, as observed using atomic force microscope (Supplementary Fig. 13), consistent with SEM in Supplementary Fig. 14. We thus correlate the presence of iron phthalocyanine with the observation of a slightly less rough copper surface:

“The rough surface of sputtered copper provides ample adsorption sites for iron phthalocyanine, and exhibits an intensified monolayer XRD signal. We also conduct atomic force microscope studies; the roughness of approximately 9 nm on the sputtered copper (Supplementary Fig. 12), while a 9 Angstrom step height, corresponding to the predicted distance Fe···3-MPA···Cu, is observed on the iron-phthalocyanine-modified copper (Supplementary Fig. 13). The Cu surface becomes smoother following molecular assembly (Supplementary Fig. 14) but keeps (111) preferred orientation, shown in Fig. 3a (the peak at 43.5°).”

a

b

Supplementary Fig. 12 | Atomic force microscope images of sputtered copper on flat Si wafer.

(a) The scanning surface and (b) the 3D image of sputtered copper on flat Si wafer

2. The authors conducted in-situ Fe K-edge XAS for Cu-FeSA to identify the formation of metallic Fe-Cu bond, but the Cu state has not been investigated. The authors should perform in-situ Cu K-edge XAS to identify the condition of Cu.

Response: We now provide in-situ Cu K-edge XAS to identify the condition of Cu, shown in Supplementary Fig. 23. Cu retains the metallic state during CO₂RR. EXAFS shows a single Cu-Cu metallic bond without Cu-O bond, while XANES suggests the oscillation behavior of metallic Cu:

“We also investigated Cu K-edge XAS to identify the condition of Cu: Cu keeps the metallic state during CO₂RR (Supplementary Figs. 23 & 24).”

Supplementary Fig. 23 | In-situ X-ray absorption spectroscopy of Cu K-edge operating in flow cell at various current. (a) In-situ extended X-ray absorption fine structure (EXAFS) of Cu K-edge, showing a single Cu-Cu metallic bond without Cu-O bond. **(b)** In-situ X-ray absorption near-edge structure (XANES) of Cu K-edge. XANES suggests the oscillation behavior of metallic Cu.

3. The reviewer notices the valence of Fe changes significantly. The authors should analyze the valence states of Cu and Fe during CO₂RR, respectively. The authors should also provide the profiles of standard samples to identify their valence states.

Response: We now deepen the analysis and discussion of the valence states of Cu and Fe during CO₂RR. We used Fe and Fe₂O₃ as standard samples to identify the oxidation state of 2.44 for Fe before CO₂RR and approximately 0 (metallic) for Fe after CO₂RR. We employed Cu and Cu₂O as the standard samples to identify the oxidation state of 0.2 for Cu before CO₂RR and approximately 0 (metallic) for Cu after CO₂RR. We now write:

“Accordingly, the oxidation state also changes from that of a cation to a metallic one, and the electronic state differs from that of pure Fe metal (Fig. 3d and Supplementary Figs. 18-20).”

“We also investigated Cu K-edge XAS to identify the condition of Cu: Cu keeps the metallic state during CO₂RR (Supplementary Figs. 23 & 24).”

Supplementary Fig. 19 | Identification of oxidation states of Fe operating in flow cell at various current. (a) X-ray absorption near edge spectroscopy of standard samples. **(b)** Oxidation states of Fe in flow cell at various current. We use energy of the 0.5 of normalized intensity as the energy position for the oxidation states rather than the inflection point due to the pre-edge feature in the spectrum.

Supplementary Fig. 24 | Identification of oxidation states of Cu operating in flow cell at various current. (a) X-ray absorption near edge spectroscopy of standard samples. **(b)** Oxidation states of Cu in flow cell at various current. We use the first inflection point of the spectra as the energy position for identifying the oxidation states.

REVIEWER COMMENTS

Reviewer #2 (Remarks to the Author):

The authors have addressed my questions. I support the publication of this manuscript in Nature Communications in the current form.

Reviewer #3 (Remarks to the Author):
see attached

The revised manuscript pretty much addresses all of my concerns. This manuscript could be publishable if the following comment can be further clarified.

The metallic Fe resulted from the negative-potential induced demetallation process out of phthalocyanine (Pc) moiety, and subsequently diffuses to Cu(111) surface to form Cu-FeSA. Since the 3-MPA bonded FePc layer exists before the demetallation step, Fe is presumptively forced to populate on the binding sites not occupied by the thiol groups of 3-MPA. Therefore, the formation of Cu-FeSA catalytic sites may not be comparable with the scenario of simply positioning a Fe atom on Cu(111) surface, being computationally described in the current DFT simulations. Given the context of the GDE preparation, the 3-MPA-bonded Pc layer is not removed after Cu-FeSA formation, could this residual organic layer play any role to the observed high FE of CH₄ generation? The authors may want to provide a control experiment (or other justification) to elaborate the effect of 3-MPA-Pc layer.

The authors claimed the disappearance of 740 cm⁻¹ signal at high negative voltage during CO₂RR for presenting the desorption of 3-MPA from copper surface. Such a desorption process of 3-MPA should be proportionally assisted by the negative electrode potential. With comparing Figure 3f (Cu-FeSA) and Figure S32, there are confusing features identified in these spectra, particularly at ~650 and 740 cm⁻¹ regions.

(1) Strong 650 cm⁻¹ signal is shown in Figure 3f but weak corresponding feature is shown in Figure S32. How is the intensity scale defined in Figure 3f and Figure S32 becomes very important for these interpretations. It should be clearly noted in the captions.

(2) If the 740 cm⁻¹ signal correctly represents the presence of 3-MPA, the color evolution in respect to the voltage seems to be inconsistent. Noticeable red spots only appear at -0.7 ~ -1.0 V (vs RHE). Shouldn't it be gradually decayed as the voltage goes to more negative, meaning more red spots should be seen at $V > -0.7$?

If 3-MPA-Pc only plays the role as the material template for synthesizing Cu-FeSA (and no role to the catalytic mechanism as noted in the manuscript), it would also show the same superior catalytic activity with removing the 3-MPA layer before starting the catalytic measurement. With that, I would like to request the evidence of the control experiment.

Manuscript ID: NCOMMS-21-10097-A

“A Metal-Supported Single-Atom Catalytic Site Enables Carbon Dioxide Hydrogenation”

Reviewer #3 (Remarks to the Author):

The revised manuscript pretty much addresses all of my concerns. This manuscript could be publishable if the following comment can be further clarified.

The metallic Fe resulted from the negative-potential induced demetallation process out of phthalocyanine (Pc) moiety, and subsequently diffuses to Cu(111) surface to form Cu-FeSA. Since the 3-MPA bonded FePc layer exists before the demetallation step, Fe is presumptively forced to populate on the binding sites not occupied by the thiol groups of 3-MPA. Therefore, the formation of Cu-FeSA catalytic sites may not be comparable with the scenario of simply positioning a Fe atom on Cu(111) surface, being computationally described in the current DFT simulations. Given the context of the GDE preparation, the 3-MPA-bonded Pc layer is not removed after Cu-FeSA formation, could this residual organic layer play any role to the observed high FE of CH₄ generation? The authors may want to provide a control experiment (or other justification) to elaborate the effect of 3-MPA-Pc layer.

The authors claimed the disappearance of 740 cm⁻¹ signal at high negative voltage during CO₂RR for presenting the desorption of 3-MPA from copper surface. Such a desorption process of 3-MPA should be proportionally assisted by the negative electrode potential. With comparing Figure 3f (Cu-FeSA) and Figure S32, there are confusing features identified in these spectra, particularly at ~650 and 740 cm⁻¹ regions.

(1) Strong 650 cm⁻¹ signal is shown in Figure 3f but weak corresponding feature is shown in Figure S32. How is the intensity scale defined in Figure 3f and Figure S32 becomes very important for these interpretations. It should be clearly noted in the captions.

Response: We are now more clear that the applied potential in the previous Figure RS32 was -1.2 V vs RHE, at which the signal is weak at 650 cm⁻¹. We now are explicit in providing the applied potential in the revised Figure RS32 and we add a new in-situ Raman result for the case of Cu-FeSA at -0.6 V vs. RHE to make it

clearer. We are now explicit that the intensity scale of revised Figure S32 is the same as that of Figure R3f. We also state explicitly the intensity scale in the caption of Figure R3f.

The Raman spectra of the Cu and Cu-3-MPA were obtained in a dry condition. The Raman signal was not reduced by the electrolyte, so that the intensity was similar to the in-situ spectrum of the Fe-FeSA.

We now separate the Raman results for the case of in-situ and dry conditions, and these are provided in revised Figure RS32.

Fig. R3 | Materials characterization and *in-situ* investigation of iron-phthalocyanine-modified copper and iron-single-atom-anchored copper. (a) X-ray diffraction. The inset illustrates the bonding between the Cu surface and iron phthalocyanine using 3-mercaptopropionic acid. **(b)** Extended X-ray absorption fine structure (EXAFS) of Fe K-edge for the Cu-FePc GDE. **(c)** *In-situ* EXAFS and **(d)** *in-situ* XANES of Fe K-edge for identifying Cu-FeSA during CO₂RR. **(e)** Atomic resolution transmission electron microscope images and atomic elemental mapping using EELS. Dashed circles indicate the single-atom iron. **(f)** *In-situ* Raman spectroscopy for pristine Cu and Cu-FeSA. The intensity scale is 4000 c.p.s. in the spectrum.

Supplementary Fig. R32 | Comparison of 3-MPA Raman signal between Cu-3-MPA and Cu-FeSA samples. (a) In-situ Raman spectrum of the Cu-FeSA at -0.6 V and -1.2 V vs. RHE. The intensity scale is the same as Figure 3f. (b) Raman spectrum of the as-prepared Cu and Cu-3-MPA samples. The 3-MPA Raman signal vanishes during the CO₂ reduction reaction, suggesting that 3-MPA leaves the catalytic surface. The signal of H₂Pc remains observable, from which we conclude that iron atoms are attracted to the copper surface, whereas few phthalocyanine rings remain near the copper surface.

(2) If the 740 cm^{-1} signal correctly represents the presence of 3-MPA, the color evolution in respect to the voltage seems to be inconsistent. Noticeable red spots only appear at -0.7 ~ -1.0 V (vs RHE). Shouldn't it be gradually decayed as the voltage goes to more negative, meaning more red spots should be seen at $V > -0.7$?

Response: We assign the 740 cm^{-1} signal to the presence of 3-MPA and FePc (JACS, **2019**, *141*, 5684) at the Cu surface. We consider the 740 cm^{-1} signal at -0.6 V vs RHE to arise from 3-MPA, since it is close to the Cu surface. At -0.7 ~ -1.0 V vs RHE, the noticeable red spots we assign to FePc, which moves toward the copper surface and leads to an increase in the Raman intensity as the 3-MPA detaches from the Cu surface. We also observe that the intensity at 660 cm^{-1} and 935 cm^{-1} increases from -0.6 V to -0.7 V vs. RHE and then decreases at a more negative voltage, suggesting that the phthalocyanine ring of FePc is detached from the Cu surface.

The full revised context is shown below and in the Main text:

“We associate the 740 cm^{-1} signal at -0.6 V vs. RHE with 3-MPA (Supplementary Fig. R32) since it is close to the Cu surface. At -0.7 ~ -1.0 V vs. RHE, the noticeable red spots can be attributed to FePc, which moves toward the copper surface and increases the Raman intensity as the 3-MPA detaches from the Cu surface. The intensity at 660 cm^{-1} and 935 cm^{-1} increases from -0.6 V to -0.7 V vs. RHE and then decreases at a more negative voltage, suggesting that the phthalocyanine ring of FePc is detached from the Cu surface.”

(3) If 3-MPA-Pc only plays the role as the material template for synthesizing Cu-FeSA (and no role to the catalytic mechanism as noted in the manuscript), it would also show the same superior catalytic activity with removing the 3-MPA layer before starting the catalytic measurement. With that, I would like to request the evidence of the control experiment.

Response: We disassembled the flow cell and washed the gas-diffusion electrode as the iron-single-atom formed, attempting to remove the 3-MPA and H₂Pc layer. Unfortunately, we observed mainly hydrogen evolution. We propose that iron is oxidized when we disrupt the reductive current, and that the copper-supported iron-decorated surface is thus disrupted.

We sought another route to see whether the remaining 3-MPA and H₂Pc on the copper surface influence the catalytic activity. We obtained an in-situ time-evolution Raman spectrum of Cu-FeSA. Initially, we observed a signal associated with phthalocyanine ligand and 3-MPA remaining on the catalytic surface. The phthalocyanine ligand and 3-MPA then progressively depart the catalytic surface. We correlate this to the activation process in the stability measurement of Figure 4c. We conclude that the remaining 3-MPA and H₂Pc affect activity, but can be removed through the electrolyte flowing during CO₂RR.

The full revised text is shown below and is now in the Main text:

“We observe a signal associated with the phthalocyanine ligand that remains following CO₂RR, consistent with the view that iron atoms are attracted to the copper surface, and that some phthalocyanine rings remain near the copper surface but depart from the catalytic surface after approximately one hour. (Supplementary Fig. 33)”

“We offer that the activation process correlates with the fact that the phthalocyanine rings remain and then leave the copper surface, shown in Supplementary Fig. 33.”

Supplementary Fig. R33 | *In-situ* time-evolution Raman spectrum of Cu-FeSA at -1.2 V vs RHE to identify the remaining phthalocyanine rings.

REVIEWERS' COMMENTS

Reviewer #3 (Remarks to the Author):

MY concerns are fully addressed. This current form of the manuscript is ready for publication.